# DUAL-FORECASTER: A MULTIMODAL TIME SERIES MODEL INTEGRATING TEXTUAL CUES VIA DUAL-SCALE ALIGNMENT

## ABSTRACT

Time series forecasting plays a vital role for decision-making across a wide range of real-world domains, which has been extensively studied. Most existing single-modal time series models rely solely on numerical series, which suffer from the limitations imposed by insufficient information. Recent studies have revealed that multimodal models can address the core issue by integrating textual information. However, these models primarily employ coarse-grained meta information designed for the whole dataset (*e.g.*, task instruction, domain description, data statistics, etc.), while the use of sample-specific textual contexts remains underexplored. To this end, we propose Dual-Forecaster, a pioneering multimodal time series model that utilizes finer-grained textual information at the sample level through the well-designed dual-scale alignment technique. Specifically, we decouple the learning of semantic and patch-level features, enabling the direct extraction of both global semantic representations critical for cross-modal understanding and local patch features essential for time series forecasting. Our comprehensive evaluations demonstrate that Dual-Forecaster is a distinctly effective multimodal time series model that outperforms or is comparable to other state-of-the-art models, highlighting the superiority of integrating textual information for time series forecasting. This work opens new avenues in the integration of textual information with numerical time series data for multimodal time series analysis.

## 1 INTRODUCTION

With the massive accumulation of time series data in such diverse domains as retail (Leonard, 2001), electricity (Liu et al., 2023a), traffic (Shao et al., 2022), finance (Li et al., 2022), and healthcare (Kaushik et al., 2020), time series forecasting has become a key part of decision-making. To date, while extensive research has been dedicated to time series forecasting, resulting in a multitude of proposed methodologies (Hyndman et al., 2008; Nie et al., 2023; Liu et al., 2023b; Ansari et al., 2024; Zhou et al., 2023), they are predominantly confined to single-modal models that rely exclusively on numerical time series data. Recent studies have shown that simple linear models ((Zeng et al., 2023; Xu et al., 2023)) can often match or even surpass the performance of state-of-the-art (SOTA) complex models, suggesting that current single-modal approaches may be nearing a saturation point.

To improve the model's forecasting performance, it is crucial to introduce supplementary information that is not present in time series data. For example, when forecasting future product sales, combining numerical historical sales data with external factors, such as product iteration plans, strategic sales initiatives, and unforeseeable events such as pandemics, enables us to give a sales forecast that aligns more closely with business expectations. This supplementary information typically appears in the form of unstructured text, which is rich in semantic details reflecting temporal causality and system dynamics. However, quantifying such valuable information into auxiliary time series data remains challenging, which presents a significant hurdle to its integration in enhancing the reliability of time series forecasting.

Recently, there has been a surge in research proposing multimodal time series models that integrate text as an auxiliary input modality (Liu et al., 2024b; Jin et al., 2024; Liu et al., 2024a; 2025; Xu

et al., 2024; Wang et al., 2025). This methodology effectively overcomes the intrinsic limitations of traditional time series forecasting methods, thereby significantly enhancing models' accuracy and effectiveness. However, in these works, the textual input consists of coarse-grained dataset-level information, such as task instructions and dataset descriptions, making it difficult to provide finer-grained sample-level discernibility. Moreover, these models focus on aligning patch-level time series features with text for time series forecasting while overlooking the significance of semantic-level features in enhancing multimodal understanding, which hinders their ability to capture complex connections between textual and time series data. Thus, it is necessary to develop an effective alignment technique tailored for finer-grained sample-level textual data to learn multimodal embeddings that will in turn enhance forecasting.

To tackle the aforementioned challenges, we introduce Dual-Forecaster, a cutting-edge time series forecasting model. Built upon a sophisticated framework, it effectively aligns finer-grained textual data at the sample level with time series data through the meticulous-designed dual-scale alignment technique. It should be noted that the word 'Dual' in Dual-Forecaster has two different levels of meaning. On the one hand, it represents that Dual-Forecaster is a multimodal time series model capable of concurrently processing both textual and time series data. On the other hand, it denotes the model's capacity to extract features at both the semantic and patch levels, enabling a hierarchical integration of high-level semantic insights and fine-grained local patterns. Specifically, Dual-Forecaster comprises the textual branch and the temporal branch. The textual branch is designed to parse textual data and extract valuable insights embedded within, while the temporal branch specializes in modeling time series dynamics. To generate high-quality embeddings for accurate forecasting, we jointly optimize two core tasks: multimodal comprehension and time series forecasting. Central to our design is the dual-scale alignment technique, which decouples the learning of semantic and patch-level features. This enables the model to directly extract (1) global semantic representations–critical for multimodal comprehension and regularized by the **Text-Time Series Contrastive Loss**–and (2) local patch features–essential for time series forecasting, derived through the **Modality Interaction Module**.

To prove the effectiveness of our model, we conduct extensive experiments on twelve multimodal time series datasets, which consist of six constructed datasets including a synthetic dataset and five captioned public datasets, and six existing multimodal datasets. Experimental results demonstrate that Dual-Forecaster achieves competitive or superior performance when compared to other SOTA models on all datasets. In addition, ablation studies emphasize that performance improvement is attributed to the supplementary information provided by the textual data.

Our main contributions in this work are threefold:

(1) We propose a sophisticated framework for integrating textual and time series data, grounded in our dual-scale alignment technique. This framework is designed to generate time series embeddings enriched with semantic insights from text, thereby enabling Dual-Forecaster to achieve more robust forecasting performance.

(2) We introduce Dual-Forecaster, a novel time series forecasting model that tackles the critical challenge of underutilized finer-grained textual signals in multimodal time series forecasting. By systematically integrating sample-specific textual semantics with temporal dynamics, our model enables the discerning of complex inter-variable dependencies.

(3) Extensive experiments across multiple datasets validate that Dual-Forecaster achieves SOTA performance on time series forecasting task. Ablation studies further highlight the critical role of the dual-scale alignment technique, demonstrating its indispensable contribution to the model's superior performance.

## 2 RELATED WORK

**Time series forecasting.** Time series forecasting models can be roughly categorized into statistical models and deep learning models. Statistical models such as ETS, ARIMA (Hyndman et al., 2008) can be fitted to a single time series and used to make predictions of future observations. Deep learning models, ranging from the classical LSTM (Hochreiter, 1997), TCN (Bai et al., 2018), to recently popular transformer-based models (Nie et al., 2023; Zhou et al., 2022; Zhang & Yan, 2023; Liu et al., 2023b), are developed for capturing nonlinear, long-term temporal dependencies. Even

though excellent performance has been achieved on specific tasks, these models lack generalizability to diverse time series data.

To overcome the challenge, the development of pre-trained time series foundation models has emerged as a burgeoning area of research. In the past two years, several time series foundation models have been introduced (Ansari et al., 2024; Garza & Mergenthaler-Canseco, 2023; Rasul et al., 2023; Das et al., 2024; Woo et al., 2024). All of them are pre-trained transformer-based models trained on a large corpus of time series data with time-series-specific designs in terms of time features, time series tokenizers, distribution heads, and data augmentation, among others. These pre-trained time series foundation models can adapt to new datasets and tasks without extensive from-scratch retraining, demonstrating superior zero-shot forecasting capability. Furthermore, benefiting from the impressive capabilities of pattern recognition, reasoning and generalization of Large Language Models (LLMs), recent studies have further explored tailoring LLMs for time series data through techniques such as fine-tuning (Zhou et al., 2023; Xue & Salim, 2023; Gruver et al., 2024) and model reprogramming (Jin et al., 2024; Cao et al., 2024; Pan et al., 2024; Sun et al., 2023). However, existing time series forecasting models have encountered a plateau in performance due to limited information contained in time series data. There is an evident need for additional data beyond the scope of time series to further refine forecasts.

**Text-guided time series forecasting**. Some works have attempted to address the prevalent issue of information insufficiency in the manner of text-guided time series forecasting, which includes text as an auxiliary input modality. A line of work investigate how to use some declarative prompts (*e.g.*, date information, task instructions, domain expert knowledge, event description, etc.) enriching the input time series to guide LLM reasoning (Liu et al., 2024b; Jin et al., 2024; Wang et al., 2025; Liu et al., 2024c; Williams et al., 2024). These approaches fall into two categories: directly prompting LLM for time series forecasting or aligning text and time series within the language space to exploit the inference potential of the LLMs. However, they ignore the key role played by the local temporal features in time series forecasting.

An alternative text-guided time series forecasting approach is to process textual and time series data separately by using different models, and then merge the information of two modalities through a modality interaction module to yield enriched time series representations for time series forecasing (Liu et al., 2024a; 2025; Xu et al., 2024). Our method belongs to this category, however, there is limited relevant research on time series. Liu et al. (2024a) develops MM-TFSlib, which provides a convenient multimodal integration framework. It can independently model numerical and textual series using different time series forecasting models and LLMs, and then combine these outputs using a learnable linear weighting mechanism to produce the final predictions. Liu et al. (2025) presents an LLM-empowered framework via cross-modality alignment for multivariate time series forecasting. The cross-modality alignment module aggregates the time series and LLM branches based on channel-wise similarity retrieval to enhance forecasting. Distinct from these methods, we focus on investigating how to utilize finer-grained textual information at the sample level to assist in time series forecasting. Moreover, We recognize that features at different scales play unique roles in enabling multimodal time series forecasting models to achieve better multimodal understanding and more accurate time series forecasting. To this end, we propose Dual-Forecaster, which can jointly optimize semantic and patch-level features based on the dual-scale alignment technique to obtain time series representations with rich semantics, aiming to better enhance the model's time series forecasting ability.

## 3 METHODOLOGY

### 3.1 PROBLEM FORMULATION

Given a dataset of $N$ numerical time series and their corresponding textual series, $\mathcal{D} = \{(\boldsymbol{X}_{t-L:t}^{(i)}, \boldsymbol{X}_{t:t+h}^{(i)}, \boldsymbol{S}_{t-L:t}^{(i)})\}_{i=1}^{N}$, where $\boldsymbol{X}_{t-L:t}^{(i)}$ is the input variable of the numerical time series, $L$ is the specified *look back window* length, and $\boldsymbol{X}_{t:t+h}^{(i)}$ is the ground truth of *horizon window* length $h$. $\boldsymbol{S}_{t-L:t}^{(i)}$ is the overall description of $\boldsymbol{X}_{t-L:t}^{(i)}$, which can be used to augment the model's capacity to learn the relationships between different time series by combining detailed descriptive information about the time series. The goal is to maximize the log-likelihood of the pre-

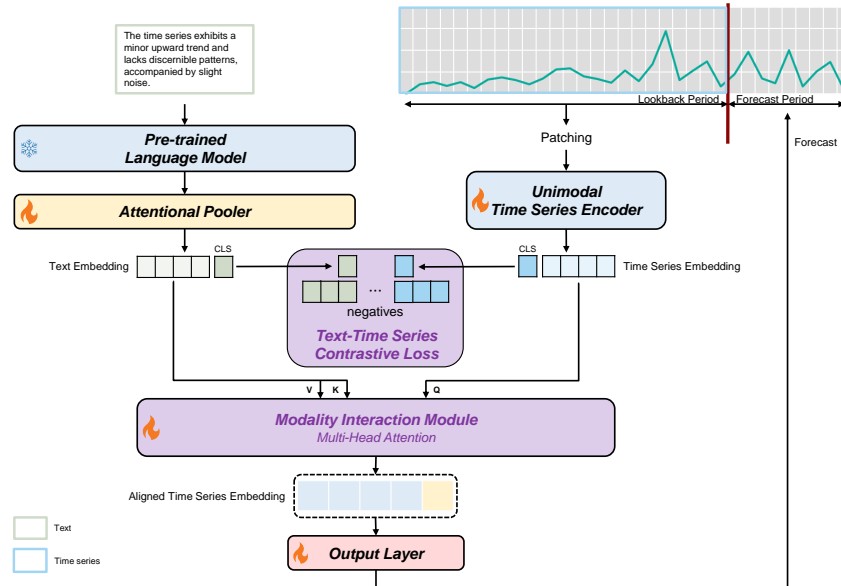

Figure 1: Overall architecture of Dual-Forecaster. Top left is the textual branch with text as input, and top right is the temporal branch with time series as input. Based on the obtained textual features and time series embeddings, to achieve effective alignment of semantic and patch-level features, we propose the **Dual-Scale Alignment** framework that employs the *Text-Time Series Contrastive Loss* and the *Modality Interaction Module*. The outputs of time series embeddings from the *Modality Interaction Module* are then projected through the Output Layer to generate the final forecasts.

dicted distribution $p\left(X_{t:t+h}|\hat{\phi}\right)$ obtained from the distribution parameters $\hat{\phi}$ learned by the model $f_{\theta} : (X_{t-L:t}, S_{t-L:t}) \rightarrow \hat{\phi}$ based on historical time series data and its corresponding descriptive textual information.

$$\max_{\theta} \mathbb{E}_{(X,S)\sim p(\mathcal{D})} \log p\left(X_{t:t+h}|\hat{\phi}\right)$$
$$s.t.\hat{\phi} = f_{\theta} : (X_{t-L:t}, S_{t-L:t}) \tag{1}$$

where $p(\mathcal{D})$ is the data distribution used for sampling numerical time series and their corresponding textual series.

## 3.2 ARCHITECTURE

Illustrated in Figure 1, our proposed Dual-Forecaster consists of two branches: the textual branch and the temporal branch. The textual branch comprises a pre-trained language model and an attentional pooler. The frozen pre-trained language model is responsible for tokenization, encoding, and embedding of text. The attentional pooler is adopted to customize textual representations produced by the language model into different scales for two core tasks of multimodal understanding and time series forecasting. The temporal branch consists of a unimodal time series encoder that is used for patching and embedding of time series. It is noteworthy that the [CLS] as a global representation of time series is introduced into the embedded representation vector. In concrete, the textual branch takes the historical text $S_{t-L:t}$ as input to obtain their corresponding embeddings $\widetilde{S}_{q(t-L:t)}$, $\widetilde{S}_{CLS(t-L:t)}$. The temporal branch works with the historical time series $X_{t-L:t}$ to obtain its corresponding embedding $\widetilde{X}_{P(t-L:t)}$, $\widetilde{X}_{CLS(t-L:t)}$. To achieve effective alignment of semantic-level features ($\widetilde{S}_{CLS(t-L:t)}$, $\widetilde{X}_{CLS(t-L:t)}$) and patch-level features ($\widetilde{S}_{q(t-L:t)}$, $\widetilde{X}_{P(t-L:t)}$), we implement the dual-scale alignment technique, which is composed of two key components: the text-time series contrastive loss and the modality interaction module. In the following section, we will provide a detailed explanation of these two components.

### 3.2.1 TEXT-TIME SERIES CONTRASTIVE LOSS

Previous multimodal time series models, whether they integrate historical texts like descriptions of input time series (Liu et al., 2024a; 2025) or future texts such as news and channel descriptions (Xu et al., 2024), typically utilize separate textual and temporal branches to process their respective modality data. Subsequently, they employ a cross-attention-based modality interaction module to facilitate the integration of these distinct modality data. Given that the textual features and time series embeddings reside in their own high-dimensional spaces, it is challenging for these models to effectively learn and model their interactions. While model reprogramming techniques like Time-LLM (Jin et al., 2024) align time series representations into the language space, thus unleashing the potential of LLM as a predictor, these approaches often overlook the critical role of local temporal features.

Inspired by the VLP framework in CV (Li et al., 2021; Yu et al., 2022), in this work, we attempt to align textual features and time series embeddings into the unified high-dimensional space before fusing in the modality interaction modules. Therefore, we develop the text-time series contrastive loss to deal with this problem. Specifically, for each input time series $\boldsymbol{X}_{t-L:t}^{(i)} \in \mathbb{R}^{1 \times L}$, it is first normalized to have zero mean and unit standard deviation in mitigating the time series distribution shift. Then, we divide it into $P$ consecutive non-overlapping patches with length $L_p$. Given these patches $\boldsymbol{X}_{P(t-L:t)}^{(i)} \in \mathbb{R}^{P \times L_p}$, we adopt a simple linear layer to embed them as $\hat{\boldsymbol{X}}_{P(t-L:t)}^{(i)} \in \mathbb{R}^{P \times d_m}$, where $d_m$ is the dimensions of time series features. On this basis, we introduce the time series CLS token $\hat{\boldsymbol{X}}_{CLS(t-L:t)}^{(i)} \in \mathbb{R}^{1 \times d_m}$. Let $\hat{\boldsymbol{X}}_{t-L:t}^{(i)} = \left[ \hat{\boldsymbol{X}}_{P(t-L:t)}^{(i)} \; \hat{\boldsymbol{X}}_{CLS(t-L:t)}^{(i)} \right] \in \mathbb{R}^{(P+1) \times d_m}$. We use the $n_{uni}$ unimodal time series encoder layers containing *Multi-Head Self-Attention* (*MHSA*) layers to process time series, and finally take the outputs of the $n_{uni}^{th}$ layer as the embeddings $\widetilde{\boldsymbol{X}}_{t-L:t}^{(i)} \in \mathbb{R}^{(P+1) \times d_m}$:

$$\widetilde{\boldsymbol{X}}_{t-L:t}^{(i)} = \left( MHSA \left( \hat{\boldsymbol{X}}_{t-L:t}^{(i)} \right) + \hat{\boldsymbol{X}}_{t-L:t}^{(i)} \right)_{n_{uni}^{th}} = \left[ \widetilde{\boldsymbol{X}}_{P(t-L:t)}^{(i)} \; \widetilde{\boldsymbol{X}}_{CLS(t-L:t)}^{(i)} \right] \tag{2}$$

For each historical text $\boldsymbol{S}_{t-L:t}^{(i)}$, we use the pre-trained language model for tokenization, encoding, and embedding to obtain $\hat{\boldsymbol{S}}_{G(t-L:t)}^{(i)} \in \mathbb{R}^{G \times d}$, where $G$ represents the number of tokens encoded in the historical text and $d$ is the dimensions of textual features. On this basis, we introduce learnable text query $\hat{\boldsymbol{Q}}_q^{(i)} \in \mathbb{R}^{q \times d_m}$ and text CLS token $\hat{\boldsymbol{Q}}_{CLS}^{(i)} \in \mathbb{R}^{1 \times d_m}$. Let $\hat{\boldsymbol{Q}}^{(i)} = \left[ \hat{\boldsymbol{Q}}_q^{(i)} \; \hat{\boldsymbol{Q}}_{CLS}^{(i)} \right] \in \mathbb{R}^{(q+1) \times d_m}$. We use a *Multi-Head Cross-Attention* (*MHCA*) layer with $\hat{\boldsymbol{Q}}^{(i)}$ as query and $\hat{\boldsymbol{S}}_{G(t-L:t)}^{(i)}$ as key and value to obtain the embedding $\widetilde{\boldsymbol{S}}_{t-L:t}^{(i)} \in \mathbb{R}^{(q+1) \times d_m}$:

$$\widetilde{\boldsymbol{S}}_{t-L:t}^{(i)} = MHCA \left( \hat{\boldsymbol{Q}}^{(i)}, \hat{\boldsymbol{S}}_{G(t-L:t)}^{(i)} \right) = \left[ \widetilde{\boldsymbol{S}}_{q(t-L:t)}^{(i)} \; \widetilde{\boldsymbol{S}}_{CLS(t-L:t)}^{(i)} \right] \tag{3}$$

Given the outputs of $\widetilde{\boldsymbol{S}}_{CLS(t-L:t)}^{(i)}$ and $\widetilde{\boldsymbol{X}}_{CLS(t-L:t)}^{(i)}$ from the textual branch and the temporal branch, respectively, the text-time series contrastive loss is defined as:

$$\text{sim}_i = \widetilde{\boldsymbol{X}}_{CLS(t-L:t)}^{(i)} \bigodot \widetilde{\boldsymbol{S}}_{CLS(t-L:t)}^{(i)}$$

$$\mathcal{L}_{contrastive} = -\frac{1}{B} \left( \sum_i^B \log \frac{exp \left( \text{sim}_i^T y_i / \tau \right)}{\sum_{j=1}^B exp \left( \text{sim}_j^T y_j \right)} + \sum_i^B \log \frac{exp \left( y_i^T \text{sim}_i / \tau \right)}{\sum_{j=1}^B exp \left( y_j^T \text{sim}_j \right)} \right) \tag{4}$$

where $B$ is the batch size, $y_i \in \mathbb{R}^{B \times B}$ is the one-hot label matrix from ground truth text-time series pair label, and $\tau$ is the temperature to scale the logits.

### 3.2.2 MODALITY INTERACTION MODULE

To ensure effective alignment of distributions between historical textual and time series data, we use the $n_{mul}$ multimodal layers, including a *MHSA* operation and a *MHCA* operation in each layer, as the

modality interaction module to obtain the aligned time series embedding that integrates historical textual information. Formally, given $\widetilde{S}_{q(t-L:t)}^{(i)}$ and $\widetilde{X}_{P(t-L:t)}^{(i)}$ produced by the textual branch and the temporal branch, at each layer of the modality interaction module, we sequentially process and aggregate the textual and temporal information based on the *MHSA* and *MHCA* mechanism, and finally take the outputs of the $n_{mul}^{th}$ layer as the aligned time series embeddings $\bar{X}_{align}^{(i)} \in \mathbb{R}^{P \times d_m}$:

$$\bar{X}_{align}^{(i)} = \left( MHCA \left( MHSA \left( \widetilde{X}_{P(t-L:t)}^{(i)} \right) + \widetilde{X}_{P(t-L:t)}^{(i)}, \widetilde{S}_{q(t-L:t)}^{(i)} \right) + \widetilde{X}_{P(t-L:t)}^{(i)} \right)_{n_{mul}^{th}} \tag{5}$$

### 3.2.3 OUTPUT LAYER

To maintain homogeneity with $\mathcal{L}_{contrastive}$, we use negative log-likelihood loss as the forecast loss, which constrains the model's predicted distribution to closely align with the actual distribution. Specifically, given $\bar{X}_{align}^{(i)}$, we linearly project the last token embedding $\bar{X}_{align[-1]}^{(i)} \in \mathbb{R}^{1 \times d_m}$ to obtain the distribution parameters of the Student's T-distribution prediction head. The forecast loss used is defined as:

$$\mathcal{L}_{forecast} = -\frac{1}{B} \sum_i^B \log p \left( X_{t:t+h}^{(i)} | \hat{\phi} \left( \bar{X}_{align[-1]}^{(i)} \right) \right) \tag{6}$$

The overall loss during training is the summation of the forecast loss $\mathcal{L}_{forecast}$ and the contrastive loss $\mathcal{L}_{contrastive}$ as follows:

$$\mathcal{L} = \mathcal{L}_{forecast} + \mathcal{L}_{contrastive} \tag{7}$$

## 4 MAIN RESULTS

**Datasets.** To demonstrate the effectiveness of the proposed Dual-Forecaster, we employ three types of dataset—synthetic dataset, captioned-public dataset, and existing multimodal time series dataset—encompassing a spectrum of difficulty levels from simple to complex. These three dataset categories exhibit different degrees of authenticity: synthetic dataset consists of synthetic time series-text pairs; captioned-public dataset contains real time series data with corresponding synthetic text annotations, and existing multimodal time series dataset are more reflective of real-world scenarios. Leveraging these datasets enables a systematic, stepwise validation of the practical effectiveness of the Dual-Forecaster. We firstly design six multimodal time series benchmark datasets across two categories: synthetic dataset and captioned-public dataset. Additionally, we gather six existing multimodal time series benchmark datasets from the Time-MMD dataset. The Time-MMD dataset (Liu et al., 2024a) encompasses such nine primary data domains as climate, health, energy, and traffic. It is the first high-quality and multi-domain, multimodal time series dataset, providing great convenience for verifying the model's multimodal time series forecasting ability in real-world scenarios. We conduct extensive experiments on them and compare Dual-Forecaster against a collection of representative methods from the recent time series forecasting landscape, our approach displays competitive or stronger results in multiple benchmarks.

**Baseline Models.** We carefully select 11 forecasting methods as our baselines which fall into two categories: *single-modal models* and *multimodal models*. For the *single-modal models*, they include DLinear (Zeng et al., 2023), FITS (Xu et al., 2023), PatchTST (Nie et al., 2023), iTransformer (Liu et al., 2023b), and Chronos (Ansari et al., 2024). For the *Multimodal models*, they consist of GPT4TS (Zhou et al., 2023), UniTime (Liu et al., 2024b), Time-LLM (Jin et al., 2024), MM-TSFlib (Liu et al., 2024a), TimeCMA (Liu et al., 2025) and ChatTime (Wang et al., 2025). It is worth noting that we employ GPT2 as LLM backbone and iTransformer as time series forecasting backbone for the MM-TSFlib model based on the experimental results reported in the original paper. We contrast Dual-Forecaster with the *single-modal models* to illustrate how textual insights can enhance forecasting performance. Comparisons with the *multimodal models* highlight the advancement of Dual-Forecaster's dual-scale alignment technique in cross-modality alignment, demonstrating its superiority over direct prompting and simple multimodal fusion methods, which in turn can further elevate the model's forecasting performance. Note that all these methods train a dedicated model for each evaluated dataset except for two foundation models of Chronos and ChatTime, which are directly used for inference.

Table 1: Forecasting result on synthetic dataset.The best and second best results are in **bold** and underlined.

| Methods | | Dual-Forecaster | | GPT4TS | | UniTime | | Time-LLM | | MM-TSFlib | | TimeCMA | | ChatTime | | Chronos | | DLinear | | FITS | | PatchTST | | iTransformer | |
|---|---|---|---|---|---|---|---|---|---|---|---|---|---|---|---|---|---|---|---|---|---|---|---|---|
| Metric | | MSE | MAE | MSE | MAE | MSE | MAE | MSE | MAE | MSE | MAE | MSE | MAE | MSE | MAE | MSE | MAE | MSE | MAE | MSE | MAE | MSE | MAE | MSE | MAE |
| synthetic dataset | 30 | **0.5970** | **0.5224** | 0.9467 | 0.7139 | 0.6684 | 0.5911 | 0.8907 | 0.6976 | 0.6013 | 0.5419 | 3.0644 | 1.4267 | 1.1251 | 0.7111 | 0.9273 | 0.6326 | 1.2190 | 0.8139 | 2.7585 | 1.3254 | 0.6015 | 0.5394 | 0.6190 | 0.5529 |

Table 2: Forecasting result on captioned-public datasets.The best result is highlighted in **bold** and the second best is highlighted in underlined.

| Methods | | Dual-Forecaster | | GPT4TS | | UniTime | | Time-LLM | | MM-TSFlib | | TimeCMA | | ChatTime | | Chronos | | DLinear | | FITS | | PatchTST | | iTransformer | |
|---|---|---|---|---|---|---|---|---|---|---|---|---|---|---|---|---|---|---|---|---|---|---|---|---|
| Metric | | MSE | MAE | MSE | MAE | MSE | MAE | MSE | MAE | MSE | MAE | MSE | MAE | MSE | MAE | MSE | MAE | MSE | MAE | MSE | MAE | MSE | MAE | MSE | MAE |
| ETTm1 | 96 | **1.3203** | **0.8061** | 1.4641 | 0.8787 | 1.4034 | 0.8607 | 1.4457 | 0.8730 | 1.3620 | 0.8426 | 2.4790 | 1.2475 | 1.7347 | 0.9613 | 1.5638 | 0.8987 | 1.5601 | 0.9198 | 2.2858 | 1.1810 | 1.4544 | 0.8619 | 1.3393 | 0.8299 |
| ETTm2 | 96 | **0.9363** | **0.6108** | 1.1184 | 0.7042 | 1.0397 | 0.6752 | 1.1199 | 0.7054 | 1.0325 | 0.6691 | 2.0336 | 1.0727 | 1.8680 | 0.9642 | 1.6991 | 0.8582 | 1.1603 | 0.7332 | 1.7418 | 0.9709 | 0.9419 | 0.6280 | 1.0210 | 0.6557 |
| ETTh1 | 96 | **1.3955** | **0.9017** | 1.5078 | 0.9495 | 1.4647 | 0.9178 | 1.5919 | 0.9914 | 1.4967 | 0.9347 | 1.6117 | 1.0065 | 1.9604 | 1.0821 | 1.5282 | 0.9402 | 1.4999 | 0.9505 | 1.6004 | 0.9952 | 1.6009 | 0.9603 | 1.5128 | 0.9438 |
| ETTh2 | 96 | **0.9429** | **0.7467** | 0.9612 | 0.7679 | 1.0028 | 0.7820 | 1.0586 | 0.8083 | 0.9616 | 0.7644 | 1.4177 | 0.9220 | 1.5014 | 0.9530 | 1.0474 | 0.7761 | 0.9951 | 0.7847 | 1.2858 | 0.8875 | 1.0349 | 0.7879 | 0.9803 | 0.7703 |
| exchange-rate | 96 | **2.2011** | **0.8458** | 3.0947 | 1.1203 | 2.5676 | 0.9933 | 3.0564 | 1.1111 | 2.6365 | 1.0061 | 4.4906 | 1.4850 | 2.3079 | 1.0291 | 2.7269 | 0.9773 | 3.1668 | 1.1146 | 4.4656 | 1.4831 | 2.2656 | 1.0016 | 2.6426 | 0.9977 |

**Implementation Details.** We utilize a six-layers pre-trained *RoBERTa* (Liu, 2019) model to process text inputs. All experiments are repeated three times. All computations are performed on a single NVIDIA GeForce RTX 4070 Ti GPU.

## 4.1 EVALUATION ON SYNTHETIC DATASET

**Setups.** The synthetic dataset is adopted to assess the model's capacity to utilize textual information for time series forecasting while effectively mitigating distribution drift. It is composed of simulated time series data containing different proportions of trend, seasonality, noise components, and switch states. For a fair comparison, the input time series *look back window* length $L$ is set as 200, and the prediction horizon $h$ is set as 30. Consistent with prior works, we choose the Mean Square Error (MSE) and Mean Absolute Error (MAE) as the default evaluation metrics.

**Results.** Table 1 presents the performance comparison of various models on synthetic dataset. Our model consistently outperforms all baseline models.

## 4.2 EVALUATION ON CAPTIONED-PUBLIC DATASETS

**Setups.** The captioned-public datasets are utilized to evaluate the model's capability of better performing time series forecasting by combining textual information to eliminate uncertainty in complex time series scenarios. They consist of the captioned version of ETTm1, ETTm2, ETTh1, ETTh2, and exchange-rate datasets which have been extensively adopted for benchmarking various time series forecasting models. In this case, the input time series *look back window* length $L$ is set to 336, and the prediction horizon $h$ is fixed as 96. It should be noted that due to resource constraints, we construct relatively small datasets on the basis of these datasets by setting the value of stride and conduct experiments on them. For ETTm1 and ETTm2 datasets, stride is set to 16, while for ETTh1 and ETTh2 datasets, stride is fixed as 4. For exchange-rate datasets, stride is set to 12.

**Results.** As demonstrated in Table 2, Dual-Forecaster consistently surpasses all baselines by a large margin, over **2.3%/4.6%** w.r.t. the second-best in MSE/MAE reduction.

## 4.3 EVALUATION ON EXISTING MULTIMODAL TIME SERIES DATASETS

**Setups.** With the increasing availability of multimodal time series datasets, we have assembled a collection of existing datasets to further validate the Dual-Forecaster's real-world applicability. These datasets from the Time-MMD datasets feature more general textual data, such as reports and news, rather than time series shape-based descriptions. Moreover, the textual data in these datasets contains varying degrees of inaccuracies, which is more in line with the real-world scenarios.

**Results.** As demonstrated in Table 3, Dual-Forecaster consistently outperforms all baselines by a significant margin, achieving a **12.5%** reduction in MSE compared to the second-best model. This underscores the actual effectiveness of Dual-Forecaster in real-world forecasting scenarios.

Table 3: Forecasting result on existing multimodal time series datasets.The best result is highlighted in **bold** and the second best is highlighted in underlined.

| Methods | | Dual-Forecaster | | GPT4TS | | UniTime | | Time-LLM | | MM-TSFlib | | TimeCMA | | ChatTime | | Chronos | | DLinear | | FITS | | PatchTST | | iTransformer | |
|---|---|---|---|---|---|---|---|---|---|---|---|---|---|---|---|---|---|---|---|---|---|---|---|---|---|
| Metric | | MSE | MAE | MSE | MAE | MSE | MAE | MSE | MAE | MSE | MAE | MSE | MAE | MSE | MAE | MSE | MAE | MSE | MAE | MSE | MAE | MSE | MAE | MSE | MAE |
| Time-MMD-Climate | 8 | **0.8520** | **0.7496** | 1.4222 | 0.9604 | 1.2687 | 0.8827 | 1.3398 | 0.9217 | 1.1386 | 0.8212 | 1.3713 | 0.9444 | 1.5118 | 0.9420 | 1.0346 | 0.8365 | 2.9057 | 1.4165 | 1.5498 | 0.9776 | 1.0975 | 0.8054 | 1.1392 | 0.8194 |
| Time-MMD-Economy | 8 | **0.1785** | **0.3366** | 0.2350 | 0.3902 | 0.2691 | 0.4116 | 0.2310 | 0.3858 | 0.2066 | 0.3583 | 0.2234 | 0.3708 | 0.3199 | 0.4491 | 0.2782 | 0.4229 | 8.1634 | 2.5406 | 0.2766 | 0.4184 | 0.1960 | 0.3512 | 0.1963 | 0.3493 |
| Time-MMD-SocialGood | 8 | **1.2364** | **0.4978** | 1.5420 | 0.6398 | 1.8239 | 0.6488 | 1.5172 | 0.6106 | 1.8615 | 0.5550 | 1.5433 | 0.6142 | 1.6290 | 0.6176 | 1.5912 | 0.6228 | 4.3273 | 1.8199 | 1.7227 | 0.6789 | 1.7509 | 0.5761 | 1.7128 | 0.5379 |
| Time-MMD-Traffic | 8 | **0.1814** | 0.2984 | 0.2763 | 0.3953 | 0.3127 | 0.4043 | 0.2299 | 0.3370 | 0.1892 | **0.2491** | 0.2805 | 0.3971 | 0.4648 | 0.5312 | 0.3920 | 0.4944 | 4.3517 | 1.8561 | 0.3542 | 0.4570 | 0.1828 | 0.2591 | 0.1917 | 0.2503 |
| Time-MMD-Energy | 12 | **0.0853** | **0.2015** | 0.2069 | 0.3445 | 0.1158 | 0.2527 | 0.1263 | 0.2619 | 0.1146 | 0.2472 | 0.2553 | 0.3872 | 0.5051 | 0.4971 | 0.1193 | 0.2297 | 1.2069 | 0.8204 | 0.2893 | 0.4086 | 0.1031 | 0.2217 | 0.1123 | 0.2415 |
| Time-MMD-Health-US | 12 | **0.8563** | **0.5931** | 1.5428 | 0.8597 | 1.0753 | 0.7117 | 1.1728 | 0.7331 | 0.9710 | 0.6308 | 2.0369 | 1.0064 | 1.5114 | 0.8102 | 1.5394 | 0.7680 | 2.2140 | 1.0484 | 2.2509 | 1.1193 | 1.1000 | 0.7233 | 1.0467 | 0.6573 |

## 4.4 MODEL ANALYSIS

**Cross-modality Alignment.** To better illustrate the effectiveness of the model design in Dual-Forecaster, we construct four model variants and conduct ablation experiments on synthetic dataset and ETTm2 dataset. The experimental results presented in Table 4 demonstrate the importance of integrating textual information for time series forecasting to achieve optimal performance, and also validate the soundness of the design of the dual-scale alignment techniques. Employing textual information results in MSE/MAE of **0.5970/0.5224** (versus 0.6135/0.5379) on synthetic dataset and **0.9363/0.6108** (versus 0.9507/0.6060) on ETTm2, respectively. Without Modality Interaction Module, we observe an average performance degradation of **0.7%**, while the average performance reduction becomes more obvious (**1.4%**) in the absence of Text-Time Series Contrastive Loss. Experimental results demonstrate that as the two components of the dual-scale alignment technique, Text-Time Series Contrastive Loss and Modality Interaction Module are both critical for deriving high-quality and semantically rich time series representations. Notably, the absence of the Text-Time Series Contrastive Loss leads to more significant performance degradation in metrics. We attribute this to Text-Time Series Contrastive Loss's discernibility at the sample level, which enables it to effectively capture inter-variable relationships and reflect them in the final time series representations.

Table 4: Ablation on synthetic dataset and ETTm2 with prediction horizon 30 and 96, respectively. The best results are highlighted in **bold**.

| Model Variants | synthetic dataset | | ETTm2 | |
|---|---|---|---|---|
| | MSE | MAE | MSE | MAE |
| **Dual-Forecaster** | **0.5970** | **0.5224** | **0.9363** | **0.6108** |
| **w/o Texts** | 0.6135 | 0.5379 | 0.9507 | 0.6060 |
| **w/ Texts** | | | | |
| → w/o Text-Time Series Contrastive Loss | 0.6057 | 0.5315 | 0.9571 | 0.6117 |
| → w/o Modality Interaction Module | 0.6038 | 0.5254 | 0.9480 | 0.6102 |

**Cross-modality Alignment Interpretation.** We present a case study on synthetic dataset, as illustrated in Figure 2, to demonstrate the alignment effect between text and time series. This is achieved by displaying the similarity matrix that captures the relationship between text features and time series embeddings. The time series data is visualized above the matrix, while its corresponding text descriptions are on the left. For example, the 6th subplot depicts a sequence with an exponential upward trend over time, corresponding to the text description "exponential upward trend". Our model accurately establishes the correlation between text and time series, as evidenced by the high similarity between their representations (the value at the 6th row and 6th column of the similarity matrix is 0.94). Additionally, the text description enables the establishment of varying degrees of associations between different variables. For instance, the similar upward trends in the 6th subplot and the 1st/4th subplots are manifested as values of 0.41 and 0.42 at the 6th row and 1st/4th columns in the similarity matrix, respectively. In contrast, less relevant variables show low values in the similarity matrix. This result shows that Dual-Forecaster is capable of autonomously discern potential connections between text and time series except be able to accurately recognize the genuine pairing text-time series relationships. This indicates that our model possesses advanced multimodal comprehension capability, which has a positive influence on improving the model's forecasting performance.

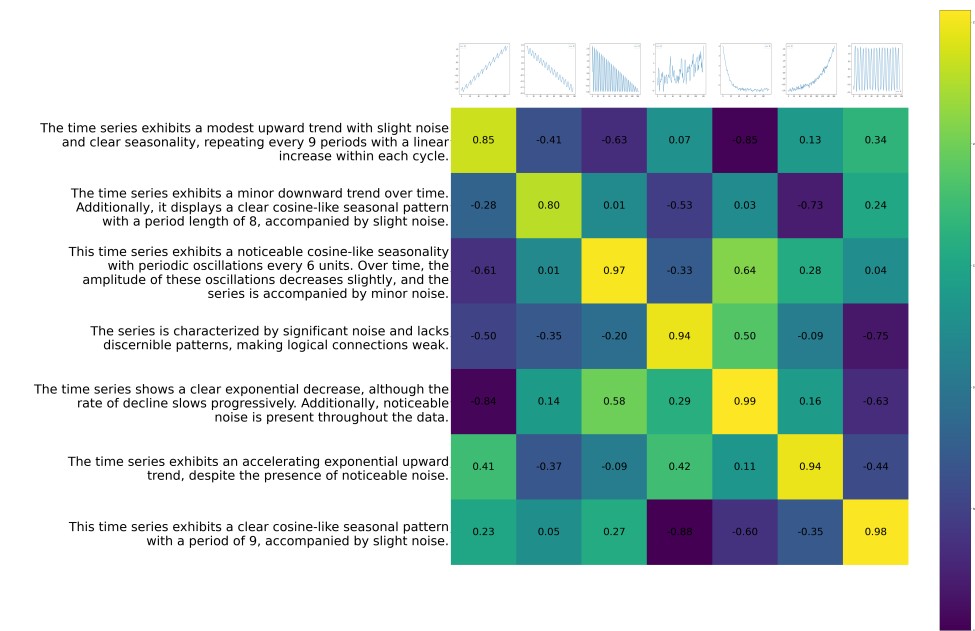

Figure 2: A showcase of text-time series alignment. The values in the matrix represent the similarity between the high-dimensional representation of the time series (above the matrix) and the corresponding textual description (on the left side of the matrix). The higher the similarity, the better the match between the time series and the text.

**Cross-modality Alignment Efficiency**   Table 5 provides an overall efficiency analysis of Dual-Forecaster with and without cross-modal alignment techniques. Our model's unimodal time series encoder is lightweight, and the overall efficiency of Dual-Forecaster is actually capped by the leveraged effective cross-modal alignment module. This is favorable in balancing forecating performance and efficiency.

## 5   CONCLUSION

In this work, we present Dual-Forecaster, an innovative multimodal time series model that integrates sample-specific textual semantics with temporal dynamic to generate more accurate and reasonable forecasts. Our model capitalizes on the meticulously-designed dual-scale alignment technique comprising the text-time series contrastive loss and modality interaction module. This technique is designed to concurrently extract semantic and patch-level features, which are crucial for cross-modal understanding and time series forecasting, respectively. We conduct extensive experiments on twelve datasets to demonstrate the effectiveness of Dual-Forecaster and highlight the superiority of incorporating textual data for time series forecasting.

**Limitations & Future Work**   While Dual-Forecaster has achieved remarkable performance in text-guided time series forecasting, there remains room for further improvements. Due to resource constraints, a comprehensive hyperparameter tuning was not performed, suggesting that the reported results of Dual-Forecaster may be sub-optimal. In terms of multimodal time series dataset, the lack of a standardized and efficient annotation methodology often leads to inadequate annotation quality on real-world datasets, with the issue being particularly pronounced in the annotation of long time series. Future work should focus on developing a more elegant time series annotator, leveraging the text-time series alignment techniques that are fundamental to Dual-Forecaster. In terms of downstream task, further research should explore the potential of expanding Dual-Forecaster to encompass a broad spectrum of multimodal time series analysis capabilities.

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

## A    Cross-modality Alignment Efficiency

Table 5: Efficiency analysis of Dual-Forecaster on synthetic dataset and ETTm2.

| Dataset-Prediction Horizon | synthetic dataset-30 | | | | ETTm2-96 | | | |
|---|---|---|---|---|---|---|---|---|
| Metric | Trainable Param. (M) | Non-trainable Param. (M) | Mem. (MiB) | Speed(s/iter) | Trainable Param. (M) | Non-trainable Param. (M) | Mem. (MiB) | Speed(s/iter) |
| w/ Texts | 13.5 | 82.1 | 1840 | 0.043 | 13.6 | 82.1 | 8812 | 0.242 |
| w/o Texts | 6.5 | 0 | 672 | 0.022 | 6.6 | 0 | 928 | 0.036 |

## B    Broader Impacts

This work introduces a groundbreaking exploration in time series forecasting—a multimodal time series forecasting model that leverages textual modality data to enhance predictive capabilities for time-series analysis. The broader impact of this research is multifaceted. By delivering high-fidelity and reliable forecasts, it empowers advanced decision-making in critical domains such as finance and healthcare, where precision is paramount. Moreover, its strong interpretability enables actionable insights for optimized resource allocation and enhanced patient care protocols. The societal implications are profound: this work establishes a novel framework for integrating complex time-series data with emerging AI technologies (e.g., LLMs), fundamentally transforming how time-series data is analyzed and utilized across diverse sectors. By bridging textual semantics and temporal dynamics, this approach paves the way for next-generation predictive models that address the growing demand for multimodal intelligence in real-world applications.

## C    Experimental Details

### C.1    Implementation

All the experiments are repeated three times with different seeds and we report the averaged results. Our model implementation is on Pytorch (Paszke et al., 2019) with all experiments conducted on a single NVIDIA GeForce RTX 4070 Ti GPU. Our detailed model configurations are in Table 6.

### C.2    Multimodal Time Series Benchmark Datasets Construction

In the realm of time series forecasting, there is a notable lack of high-quality multimodal time series benchmark datasets that combine time series data with corresponding textual series. While some studies have introduced multimodal benchmark datasets (Liu et al., 2024a; Xu et al., 2024), these datasets primarily rely on textual descriptions derived from external sources like news reports or background information. These types of textual data are often domain-specific and may not be consistently available across different time series domains, limiting their utility for building unified multimodal models. In contrast, shape-based textual descriptions of time series patterns are relatively easier to generate and can provide more structured insights. The TS-Insights dataset Zhang et al. (2023) pairs time series data with shape-based textual descriptions. However, these descriptions are based on detrended series (with seasonality removed), which may introduce bias and complicate the interpretation of the original time series data. To address these challenges, we propose six new multimodal time series benchmark datasets where textual descriptions are directly aligned with the observed patterns in the time series. The construction process for these datasets is outlined below.

#### C.2.1    Synthetic dataset

For the synthetic time series data, we firstly design three categories of components, which are then combined to generate simulated time series. The components are as follows:

- **Trend:** Linear trend, exponential trend
- **Seasonality:** Cosine, linear, exponential, M-shape, trapezoidal
- **Noise:** Gaussian noise with varying variances

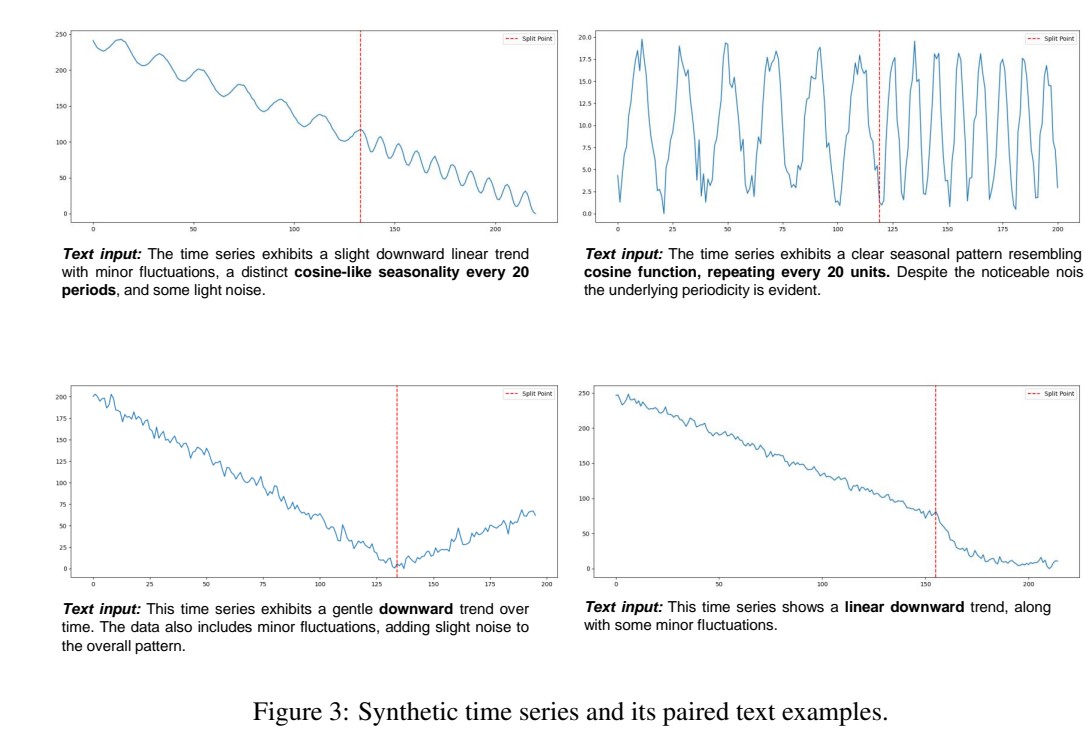

**Text input:** The time series exhibits a slight downward linear trend with minor fluctuations, a distinct **cosine-like seasonality every 20 periods**, and some light noise.

**Text input:** The time series exhibits a clear seasonal pattern resembling a **cosine function, repeating every 20 units.** Despite the noticeable noise, the underlying periodicity is evident.

**Text input:** This time series exhibits a gentle **downward** trend over time. The data also includes minor fluctuations, adding slight noise to the overall pattern.

**Text input:** This time series shows a **linear downward** trend, along with some minor fluctuations.

Figure 3: Synthetic time series and its paired text examples.

To generate the synthetic time series, one component from each category is randomly selected. These components are then either added together or multiplied to produce a time series, along with a corresponding textual description of its key characteristics. To enhance the diversity of the descriptions, rule-based descriptions are paraphrased using GPT-4o. Additionally, to simulate transitions between different states, we generate time series where only one component changes over time. For instance, a time series might exhibit a linear upward trend that transits to a linear downward trend. In this manner, we construct the synthetic dataset with a total of 3,040 training samples. Each sample includes time series and its paired textual series. Several examples of these constructed samples are shown in Figure 3.

### C.2.2 CAPTIONED PUBLIC DATASETS

For the real-world time series data, we construct corresponding textual descriptions using the following method, and Figure 4 shows the whole caption process.

- First, we apply the Iterative End Point Fitting (IEPF) algorithm (Douglas & Peucker, 1973) to the min-max normalized time series, identifying reasonable segmentation points. IEPF begins by taking the starting curve, which consists of an ordered set of points, and an allowable distance threshold. Initially, the first and last points of the curve are marked as essential. The algorithm then iteratively identifies the point farthest from the line segment connecting these endpoints. If the distance of this point exceeds threshold, it is retained as a segmentation point, and the process is recursively repeated for the subsegments until no points are found that are farther than threshold from their respective line segments. This iterative approach ensures that the segmentation preserves the curve's critical structure while discarding unnecessary details. The lines connecting these segmentation points can roughly outline the overall shape of the time series.

- Once the time series is segmented, statistical features such as slope and volatility are computed for each section. For each segment, a linear regression model is fitted to the data, and the slope is calculated. The P-value from the regression determines the significance of the trend: if it's below 0.05, the slope indicates an upward or downward trend; if it's above 0.05, the segment is considered to be fluctuating. The Mean Squared Error (MSE) between

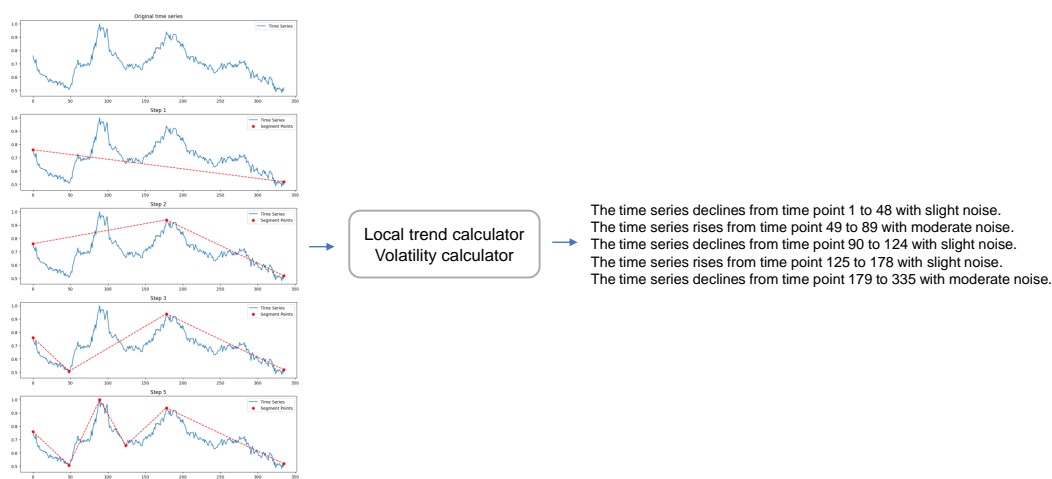

Figure 4: Captioning process for real-world time series. First, IEPF is used to segment time series, identifying reasonable segmentation points. This algorithm works by iteratively fitting straight lines between endpoints and adjusting segmentation points to minimize fitting errors, thereby identifying rational breakpoints. Next, statistical features such as slope and volatility are calculated for each segmented portion of the time series. Finally, based on these statistical characteristics, a descriptive textual summary is generated.

> the original data and the regression line is also calculated to measure the noise level. Based on the MSE, the noise is classified into three levels: low, medium, or high.

- Finally, a textual description is generated: if the local trend is significant, the description notes whether the segment is increasing or decreasing; if not, it indicates fluctuation. The noise level is also included in the description based on the MSE.

We apply the above method to annotate five commonly used real-world datasets: ETTm1, ETTm2, ETTh1, ETTh2, and exchange-rate. Each dataset is divided into training and testing sets with a ratio of 8:2. Following the configurations of a look-back window of 336 and a forecasting horizon of 96, we construct training samples using a sliding window approach. Figure 5 illustrates the text annotation results on the exchange-rate dataset. Our annotation method accurately captures segmentation points (red lines), thereby producing meaningful summary shape descriptions.

It should be noted that due to resource constraints, we construct relatively small datasets on the basis of these datasets by setting the value of stride and conduct experiments on them. For ETTm1 and ETTm2 datasets, stride is set to 16, while for ETTh1 and ETTh2 datasets, stride is fixed as 4. For exchange-rate datasets, stride is set to 12.

## C.3 MULTIMODAL TIME SERIES BENCHMARK DATASETS COLLECTION

Apart from our constructed multimodal time series datasets, including the synthetic dataset and captioned-public datasets, we also collect the Time-MMD dataset. The Time-MMD dataset (Liu et al., 2024a) encompasses such nine primary data domains as climate, health, energy, and traffic. It is the first high-quality and multi-domain, multimodal time series dataset, providing great convenience for verifying the model's multimodal time series forecasting ability in real-world scenarios.

## C.4 MODEL CONFIGURATIONS

The configurations of our models, in relation to the evaluations on various datasets, are consolidated in Table 6. By default, optimization is achieved through the Adam optimizer (Kingma, 2014) with a learning rate set at 0.0001 (0.005 for the Time-MMD-Economy dataset, Time-MMD-Energy dataset and Time-MMD-Health-US dataset) and a weight deacy ratio of 0.01, throughout all experiments. In

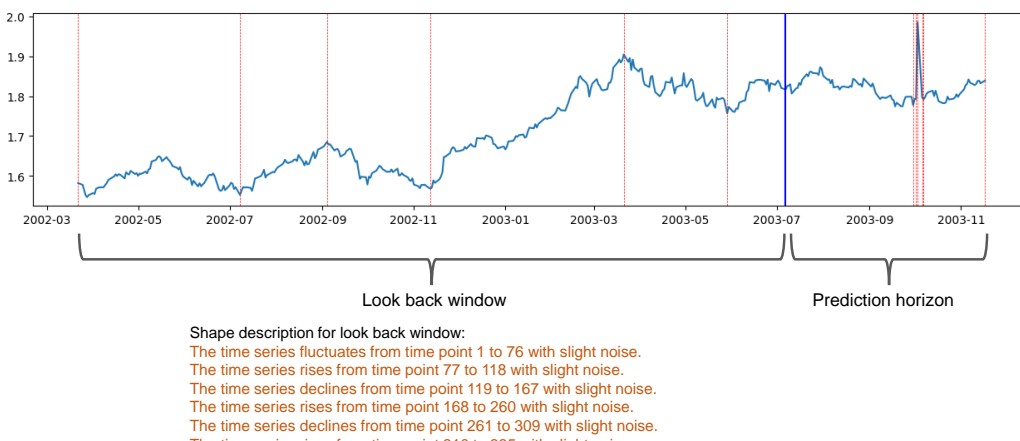

Shape description for look back window:
The time series fluctuates from time point 1 to 76 with slight noise.
The time series rises from time point 77 to 118 with slight noise.
The time series declines from time point 119 to 167 with slight noise.
The time series rises from time point 168 to 260 with slight noise.
The time series declines from time point 261 to 309 with slight noise.
The time series rises from time point 310 to 335 with slight noise.

Figure 5: Visualization of a captioned example from exchange-rate dataset.

terms of dataset parameters, $L$ and $h$ signify the input time series *look back window* length and the future time points to be predicted, respectively. For the input time series, we firstly perform patching to obtain $P$ non-overlapping patches with a patch length of $L_p$. In terms of model hyperparameters, $d_m$ represents the dimension of the embedded representations, and $n_{uni}$ denotes the number of layers of unimodal time series encoder used to process time series inputs, while $n_{mul}$ denotes the number of layers of the modality interaction module, which ensures effectively alignment of distributions between historical textual and time series data. Heads are correlate to the *Multi-Head Self-Attention* (*MHSA*) and *Multi-Head Cross-Attention* (*MHCA*) operations utilized for cross-modality alignment. For the synthetic dataset and Time-MMD datasets, we set the training epochs to 300, while for the ETT, and exchange-rate datasets, we set it to 100. Additionally, to prevent overfitting, we introduce an early stopping strategy and set the patience to 7 except for the Time-MMD-Economy dataset, Time-MMD-Energy dataset and Time-MMD-Health-US dataset.

Table 6: An overview of the experimental configurations for Dual-Forecaster.

| Dataset/Configuration | Dataset Parameter | | | | Model Hyperparameter | | | | Training Process | | | | |
|---|---|---|---|---|---|---|---|---|---|---|---|---|---|
| | $L$ | $P$ | $L_p$ | $h$ | $d_m$ | $n_{uni}$ | $n_{mul}$ | Heads | LR | Weight Decay | Batch Size | Epochs | Patience |
| synthetic dataset | 200 | 25 | 8 | 30 | 256 | 6 | 3 | 8 | 0.0001 | 0.01 | 64 | 300 | 7 |
| ETTm1 | 336 | 42 | 8 | 96 | 256 | 6 | 1 | 8 | 0.0001 | 0.01 | 64 | 100 | 7 |
| ETTm2 | 336 | 42 | 8 | 96 | 256 | 6 | 3 | 8 | 0.0001 | 0.01 | 64 | 100 | 7 |
| ETTh1 | 336 | 42 | 8 | 96 | 256 | 6 | 3 | 8 | 0.0001 | 0.01 | 64 | 100 | 7 |
| ETTh2 | 336 | 42 | 8 | 96 | 256 | 6 | 3 | 8 | 0.0001 | 0.01 | 64 | 100 | 7 |
| exchange-rate | 336 | 42 | 8 | 96 | 256 | 6 | 3 | 8 | 0.0001 | 0.01 | 64 | 100 | 7 |
| Time-MMD-Climate | 8 | 1 | 8 | 8 | 256 | 2 | 1 | 4 | 0.0001 | 0.01 | 32 | 300 | 7 |
| Time-MMD-Economy | 8 | 1 | 8 | 8 | 64 | 2 | 1 | 2 | 0.005 | 0.01 | 32 | 300 | 20 |
| Time-MMD-SocialGood | 8 | 1 | 8 | 8 | 256 | 2 | 3 | 2 | 0.0001 | 0.01 | 32 | 300 | 7 |
| Time-MMD-Traffic | 8 | 1 | 8 | 8 | 128 | 2 | 1 | 4 | 0.0001 | 0.01 | 32 | 300 | 7 |
| Time-MMD-Energy | 40 | 5 | 8 | 12 | 128 | 2 | 1 | 2 | 0.005 | 0.01 | 32 | 300 | 20 |
| Time-MMD-Health-US | 40 | 5 | 8 | 12 | 128 | 2 | 1 | 2 | 0.005 | 0.01 | 32 | 300 | 20 |

C.5 EVALUATION METRIC

We adopt the Mean Square Error (MSE) and Mean Absolute Error (MAE) as the default evaluation metrics. The calculations of these metrics are as follows:

$$MSE = \frac{1}{H}\sum_{h=1}^{H}\left(Y_h - \hat{Y}_h\right)^2$$

$$MAE = \frac{1}{H}\sum_{h=1}^{H}\left|Y_h - \hat{Y}_h\right|$$

where $H$ denotes the length of prediction horizon. $Y_h$ and $\hat{Y}_h$ are the $h$-th ground truth and prediction where $h \in \{1, \cdots, H\}$.

## D  BASELINES

**DLinear:** is a combination of a decomposition scheme and a linear network that first divides a time series data into two components of trend and remainder, and then performs forecasting to the two series respectively with two one-layer linear model.

**FITS:** consists of the key part of the complex-valued linear layer that is dedicatedly designed to learn amplitude scaling and phase shifting, thereby facilitating to extend time series segment by interpolating the frequency representation.

**PatchTST:** is composed of two key components: (i) patching that segments time series into patches as input tokens to Transformer; (ii) channel-independent structure where each channel univariate time series shares the same Transformer backbone.

**iTransformer:** is an inverted Transformer that raw series of different variates are firstly embedded to tokens, applied by self-attention for multivariate correlations, and individually processed by the share feed-forward network for series representations of each token.

**Chronos:** is a framework that adapts language model architectures and training procedures to probabilistic time series forecasting by tokenizing time series values into a fixed vocabulary.

**GPT4TS:** is a unified framework that uses a frozen pre-trained GPT2 for general time series analysis tasks including time series classification, short/long-term forecasting, imputation, anomaly detection, few-shot and zero-sample forecasting.

**UniTime:** is a unified model for cross-domain time series forecasting. It overcomes challenges like varying data characteristics, domain confusion, and convergence speed imbalance, ans shows superior performance and zero-shot transferability through experiments on multiple datasets.

**Time-LLM:** is a new framework, which encompasses reprogramming time series data into text prototype representations before feeding it into the frozen LLM and providing input context with declarative prompts via Prompt-as-Prefix to augment reasoning.

**MM-TSFlib:** is the first multimodal time-series forecasting (TSF) library, which allows the integration of any open-source language models with arbitrary TSF models, thereby enabling multimodal TSF tasks based on Time-MMD.

**TimeCMA:** is an LLM-empowered framework for multivariate time series forecasting. It addresses data entanglement issues by using a dual-modality encoding and cross-modality alignment, and reduces computational costs through last token embedding storage.

**ChatTime:** is a multimodal time series foundation model that treats time series as a foreign language. It provides zero-shot capability and supports bimodal input/output for both time series and text.

# E  ERROR BARS

All experiments are repeated three times except for ChatTime and Chronos, which execute only one inference. The comparison between our method and all the baseline methods on all datasets is delineated in Table 7.

Table 7: Standard deviations of Dual-Forecaster and baseline models across all datasets (MSE reported).

| Model | Dual-Forecaster | GPT4TS | UniTime | Time-LLM | MM-TSFlib | TimeCMA | ChatTime | Chronos | DLinear | FITS | PatchTST | iTransformer |
|---|---|---|---|---|---|---|---|---|---|---|---|---|
| Dataset | MSE | MSE | MSE | MSE | MSE | MSE | MSE | MSE | MSE | MSE | MSE | MSE |
| synthetic dataset | 0.5970±0.0182 | 0.9467±0.0570 | 0.6684±0.0241 | 0.8907±0.0296 | 0.6013±0.0020 | 3.0644±0.0132 | 1.1251 | 0.9273 | 1.2190±0.0254 | 2.7585±0.3402 | 0.6015±0.0063 | 0.6190±0.0061 |
| ETTm1 | 1.3203±0.0138 | 1.4641±0.0303 | 1.4034±0.0165 | 1.4457±0.0048 | 1.3620±0.0085 | 2.4790±0.0013 | 1.7347 | 1.5638 | 1.5601±0.0106 | 2.2858±0.1409 | 1.4544±0.0348 | 1.3393±0.0045 |
| ETTm2 | 0.9363±0.0069 | 1.1184±0.0351 | 1.0397±0.0156 | 1.1199±0.0375 | 1.0325±0.0157 | 2.0336±0.0021 | 1.8680 | 1.6991 | 1.1663±0.0060 | 1.7418±0.0415 | 0.9419±0.0040 | 1.0210±0.0123 |
| ETTh1 | 1.3955±0.0353 | 1.5078±0.0949 | 1.4647±0.0255 | 1.5919±0.1427 | 1.4967±0.0333 | 1.6117±0.0042 | 1.9604 | 1.5282 | 1.4999±0.0043 | 1.6004±0.0335 | 1.6009±0.0840 | 1.5128±0.0146 |
| ETTh2 | 0.9429±0.0332 | 0.9612±0.0183 | 1.0028±0.0312 | 1.0586±0.1008 | 0.9616±0.0080 | 1.4177±0.0070 | 1.5014 | 1.0474 | 0.9951±0.0015 | 1.2858±0.0306 | 1.0349±0.0578 | 0.9903±0.0292 |
| exchange-rate | 2.2011±0.0161 | 3.0947±0.1516 | 2.5676±0.3719 | 3.0564±0.0877 | 2.6365±0.0958 | 4.4906±0.0213 | 2.3079 | 2.7269 | 3.1668±0.0148 | 4.4656±0.2718 | 2.2656±0.0460 | 2.6426±0.0141 |
| Time-MMD-Climate | 0.8520±0.0018 | 0.14222±0.0491 | 1.2687±0.0841 | 1.3398±0.0845 | 1.1386±0.0612 | 1.3713±0.0139 | 1.5118 | 1.0346 | 2.9057±0.3899 | 1.5498±0.0868 | 1.0975±0.0130 | 1.1392±0.0186 |
| Time-MMD-Economy | 0.1785±0.0038 | 0.2350±0.0063 | 0.2691±0.0111 | 0.2310±0.0066 | 0.2066±0.0051 | 0.2234±0.0003 | 0.3199 | 0.2782 | 8.1634±1.7173 | 0.2766±0.0340 | 0.1960±0.0029 | 0.1963±0.0037 |
| Time-MMD-SocialGood | 1.2364±0.0310 | 1.5420±0.0254 | 1.8239±0.2461 | 1.5172±0.0413 | 1.8615±0.0280 | 1.5433±0.0033 | 1.6290 | 1.5912 | 4.3273±0.6212 | 1.7227±0.1214 | 1.7509±0.1066 | 1.7128±0.0585 |
| Time-MMD-Traffic | 0.1814±0.0010 | 0.2763±0.0031 | 0.3127±0.0418 | 0.2299±0.0335 | 0.1892±0.0076 | 0.2805±0.0005 | 0.4648 | 0.3920 | 4.3517±1.0442 | 0.3542±0.0761 | 0.1828±0.0026 | 0.1917±0.0017 |
| Time-MMD-Energy | 0.0853±0.0044 | 0.2069±0.0391 | 0.1158±0.0018 | 0.1263±0.0223 | 0.1146±0.0027 | 0.2553±0.0025 | 0.5051 | 0.1193 | 1.2069±0.1884 | 0.2893±0.0524 | 0.1031±0.0058 | 0.1123±0.0029 |
| Time-MMD-Health-US | 0.8563±0.0287 | 1.5428±0.1567 | 1.0753±0.0176 | 1.1728±0.0290 | 0.9710±0.1153 | 2.0369±0.0051 | 1.5114 | 1.5394 | 2.2140±0.2037 | 2.2509±0.2076 | 1.1000±0.0299 | 1.0467±0.0217 |

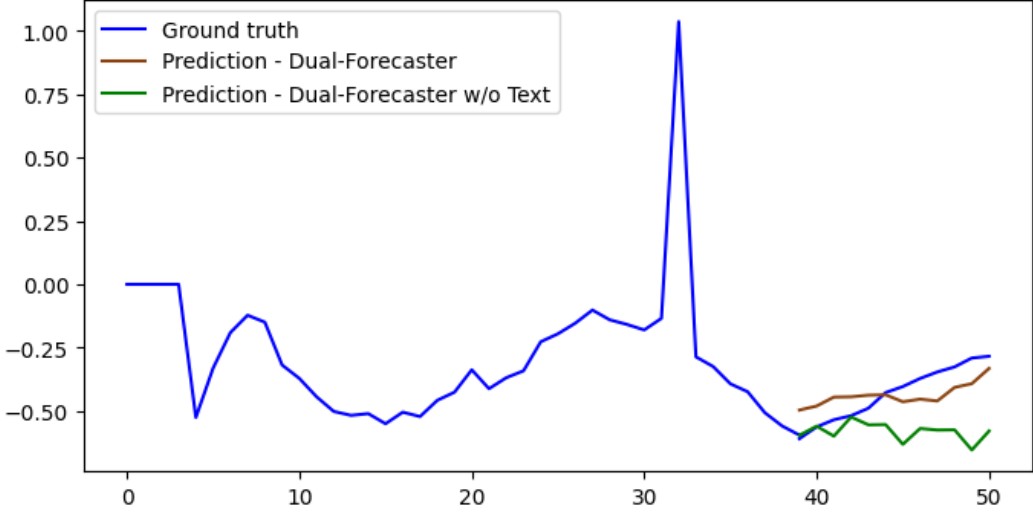

Figure 6: **Case study about interpretability of textual influence from Time-MMD-Health-US.**

