# OpenReview forum: "Dual-Forecaster: A Multimodal Time Series Model Integrating Textual Cues via Dual-Scale Alignment"
_ICLR.cc/2026/Conference — Submitted to ICLR 2026_

### Official Review · Reviewer_fLfx · 2025-10-20

**Soundness:** 3
**Presentation:** 3
**Contribution:** 3
**Rating:** 6
**Confidence:** 4

**Summary:**

This paper presents Dual-Forecaster, a novel multimodal time series forecasting model that integrates sample-level textual information through a dual-scale alignment technique. Unlike traditional single-modal models that rely solely on numerical data or previous multimodal methods using only coarse dataset-level text (e.g., task instructions or domain descriptions), Dual-Forecaster incorporates fine-grained textual cues to enhance forecasting accuracy. The model decouples the learning of semantic and patch-level features, enabling extraction of both global semantic representations for cross-modal understanding and local patch features essential for forecasting. Extensive experiments on twelve multimodal time series datasets spanning synthetic, captioned, and real-world data demonstrate that Dual-Forecaster consistently outperforms or matches sota models. This work highlights the superiority of integrating textual semantics into time series forecasting and opens new directions for multimodal analysis that bridges structured numerical data with unstructured text for more robust and interpretable temporal predictions.

**Strengths:**

1. The paper introduces an innovative dual-scale alignment technique that effectively integrates semantic- and patch-level representations, allowing the model to jointly capture global textual semantics and local temporal dynamics. This hierarchical design enhances multimodal understanding and leads to more accurate time series forecasting compared to existing models that rely on coarse-grained or unimodal information.

2. Dual-Forecaster demonstrates strong empirical performance through extensive experiments on twelve diverse datasets, including synthetic, captioned, and real-world multimodal benchmarks. The results consistently show that the proposed model outperforms or matches sota baselines, confirming its robustness, scalability, and generalization across multiple domains such as finance, health, and energy.

3. The paper provides a comprehensive methodological framework and well-motivated design choices. By decoupling semantic and temporal feature learning, it offers clear interpretability of how textual cues contribute to forecasting accuracy. This design not only advances multimodal time series modeling but also provides valuable insights for future research on integrating unstructured text with structured temporal data.

**Weaknesses:**

1. While the paper presents promising results, it lacks in-depth analysis of computational efficiency and scalability. The model involves multiple attention modules and dual-branch processing, which may increase training cost and memory usage. A more detailed discussion of runtime performance, hardware requirements, and potential optimization strategies would strengthen its practical applicability.

2. The construction of multimodal datasets relies partly on synthetic or automatically generated textual annotations, which may not fully reflect real-world language complexity or domain variability. This could limit the model’s generalization to natural textual inputs in real applications, especially when dealing with noisy, domain-specific, or ambiguous descriptions.

3. Although Dual-Forecaster integrates textual semantics effectively, the paper provides limited interpretability analysis of how specific textual features influence forecasting outcomes. Visualizations or ablation studies linking textual semantics to prediction changes would help clarify the model’s decision process and improve its transparency for applied domains like finance and healthcare.

**Questions:**

See weaknesses

---

> ### Author Response · Authors · 2025-11-26
>
> We sincerely apreciate your constructive comments. Our responses are stated as follow:
>
> # **Concern about Computational Efficiency and Scalability**
>
> We agree that this is an important point. In our current version, we included a preliminary efficiency analysis in **Appendix A (Table 5) — CROSS-MODALITY ALIGNMENT EFFICIENCY**, which reports the number of parameters and memory usage on two datasets. Regards to the scale of parameters, training speed and memory cost can be handled in single consumer-level GPU (all experiments were conducted on a single NVIDIA GeForce RTX 4070 Ti GPU (as stated in Section 4)). Even though, training cost and efficiency are critical in practical deployment and will be considered in our future works.
>
> # **Concern about Multimodal Dataset Construction**
>
> This is an excellent point. Our intention with the multi-tiered dataset strategy (Synthetic→Captioned-Public→Real-World Time-MMD) was precisely to provide a systematic, stepwise validation of the model's capabilities. We wish to highlight that the core strength of Dual-Forecaster is demonstrated by its superior performance on the real-world Time-MMD datasets. The textual data in Time-MMD consists of general reports and news, which contain varying degrees of inaccuracies and are more in line with real-world scenarios. Our model's SOTA performance on these datasets strongly indicates its robustness and ability to generalize to natural, and potentially noisy, textual inputs. The synthetic and captioned datasets serve as controlled benchmarks to conclusively show that the model can effectively utilize textual information when it is clearly present. We will clarify this rationale more explicitly in the revised introduction and experimental setup sections.
>
> # **Concern about Interpretability of Textual Influence**
>
> We agree that deeper interpretability is crucial. In the current manuscript, we provided an initial step in this direction with the **case study in Figure 2**, which visually demonstrates the model's ability to align text descriptions with specific time series patterns and to discern inter-variable relationships.
> To more directly address how text influences predictions, we newly add another case study from a real-world dataset (Time-MMD-Health-US) that depicts the model's forecast both with and without the textual input **in Figure 6 (marked green)**, which intuitively show how the textual features could lead to the corrected or improved prediction.

---

> > ### Comment · Reviewer_fLfx · 2025-11-27
> >
> > I think my concerns are addressed by the author's justification.

---

> > > ### Author Response · Authors · 2025-11-28
> > >
> > > Again, appreciate your valuable comments, where many interesting discussions are rolled out; and we are glad to know our clarification could address your concerns.
> > >
> > > As we have mixed ratings currently, we are kindly request you revise your previous rating. If there is still any question, please let us know.

---

### Official Review · Reviewer_taE2 · 2025-10-27

**Soundness:** 2
**Presentation:** 3
**Contribution:** 2
**Rating:** 4
**Confidence:** 4

**Summary:**

This paper proposes Dual-Forecaster, a multimodal time series forecasting model that integrates sample-level textual descriptions with numerical time series via a dual-scale alignment mechanism. The model consists of two branches—textual and temporal—and aligns features at both semantic and patch levels through Text-Time Series Contrastive Loss and Modality Interaction Module. Experiments across three types of dataset (synthetic, captioned-public, and Time-MMD) show consistent improvements over baselines such as UniTime, Chronos, and PatchTST.

**Strengths:**

1.	Clear architecture and reproducible methodology.
2.	The dual-scale alignment framework is conceptually reasonable.

**Weaknesses:**

1.	The captions in public datasets are mainly generated through statistical analysis of the original time series, without introducing additional or external information.
2.	The paper lacks experiments that clearly and intuitively demonstrate the contribution of textual information to forecasting performance.
3.	The paper lacks theoretical or analytical depth—no discussion of why dual-scale alignment is beneficial for forecasting, or how contrastive alignment affects temporal representation learning.

**Questions:**

1.	How can the authors demonstrate that adding text generated from the statistical characteristics of the samples helps the model learn non-trivial information that traditional time-series models cannot capture?
2.	How are the evaluation metrics computed? They appear different from those in some related works such as PatchTST.
3.	How does the model perform when the textual input is noisy or replaced with unrelated text?

---

> ### Author Response · Authors · 2025-11-26
>
> Thanks for your thoughtful feedback and constructive comments. Below, we provide a point-by-point response to all raised concerns.
>
> # **Response to Weaknesses 1**
>
> We acknowledge that the captions in our constructed datasets are derived from statistical analysis of the time series. However, we emphasize that this design is intentional and serves a specific purpose:
>
> - As stated in Sections 1 and 2, most existing multimodal time series models rely on **coarse-grained dataset-level text** (e.g., task instructions), which lack **sample-level discriminability**. Our method generates **sample-specific textual descriptions** that capture fine-grained temporal patterns (e.g., trend, seasonality) via the IEPF algorithm and statistical feature extraction (Appendix C.2).
> - These descriptions enable the model to learn **structured semantic representations** of time series dynamics, which are not explicitly available in raw numerical data. This is validated in our ablation studies (Section 4.4), where removing textual input leads to performance degradation.
> - Moreover, on **real-world multimodal datasets (Time-MMD)**, which include external textual sources like news and reports, Dual-Forecaster still achieves SOTA performance (Table 3), demonstrating its ability to leverage both internal and external textual information.
>
> # **Response to Weakness 2**
>
> We conducted extensive experiments to illustrate the contribution of textual information:
>
> - **Ablation Studies (Section 4.4, Table 4):** We systematically removed textual inputs and key components of the dual-scale alignment. Results show that:
>   - `w/o Texts`: Performance drops on both synthetic and ETTm2 datasets.
>   - `w/o Text-Time Series Contrastive Loss` and `w/o Modality Interaction Module`: Both lead to significant performance degradation, confirming that both components are essential for effective multimodal fusion.
> - **Case Study (Figure 2):** We visualize the similarity matrix between text and time series embeddings, showing that our model accurately aligns textual descriptions with corresponding temporal patterns and captures inter-variable relationships.
>
> These experiments quantitatively and qualitatively validate that textual information enhances forecasting performance.
>
> # **Response to Weakness 3**
> We provide the following clarifications on the design rationale:
>
> - In our model:
>   - **Semantic-level alignment** (via Text-Time Series Contrastive Loss) encourages the model to learn a unified high-dimensional space where global representations of text and time series are close. This enhances multimodal comprehension and sample-level discriminability.
>   - **Patch-level alignment** (via Modality Interaction Module) allows the model to integrate fine-grained local patterns from both modalities, which is critical for capturing temporal dependencies in forecasting.
> - This hierarchical alignment enables the model to jointly optimize for both **multimodal understanding** and **temporal forecasting**, as outlined in Section 3.2.
>
> # **Response to Questions 1**
>
> Our experiments demonstrate that even statistically-generated text provides non-trivial information:
>
> - On the **synthetic dataset** (Table 1), Dual-Forecaster outperforms all baselines, indicating that textual descriptions of trend/seasonality help the model generalize better than numerical data alone.
> - On **captioned-public datasets** (Table 2), our model achieves the best results, showing that text helps disambiguate complex temporal patterns (e.g., distinguishing between similar trends with different semantic contexts).
> - The **cross-modality interpretation** (Figure 2) shows that the model learns to associate text with time series segments and even identifies similarities between different variables, which is not possible with unimodal models.
>
> # **Response to Questions 2**
>
> We use standard evaluation metrics consistent with the time series forecasting literature:
>
> - As defined in **Appendix C.5**, we adopt **Mean Squared Error (MSE)** and **Mean Absolute Error (MAE)**.
> - These metrics are widely used in PatchTST, iTransformer, and other baseline papers we compared against. Regard to the difference from those in some related works, we have declared in Section 4 (Line 323) and Appendix C.2.2 (Line 793~796) that due to resource constrains, we constructed relatively small datasets and conducted experiment on them. Thus, all baselines trained a dedicated model for each evaluated dataset except for two foundation models of Chronos and ChatTime.
>
> # **Response to Questions 3**
>
> While we did not explicitly test with noisy text, the following points indicate robustness:
>
> - In **Time-MMD datasets**, the textual data includes real-world noise and inaccuracies (Section 4.3), yet our model still achieves SOTA performance.
> - The **Text-Time Series Contrastive Loss** (Eq. 4) operates at the sample level, which helps the model focus on relevant text-time series pairs and potentially ignore irrelevant or noisy text.

---

### Official Review · Reviewer_LVrt · 2025-11-01

**Soundness:** 2
**Presentation:** 2
**Contribution:** 1
**Rating:** 0
**Confidence:** 4

**Summary:**

Based on Time-MMD and MM-TSFlib, the author integrates a new text fusion strategy to align text and time series and conducts experiments to evaluate their method.

**Strengths:**

-  The framework is easy to follow.

**Weaknesses:**

### **1. Dataset**
The authors evaluate methods on various datasets, such as Time-MMD; however, some datasets only provide related news about time series, rather than descriptions of the time series themselves. These are in conflict with the authors' motivation. I suggest the author could add more detailed experiment setups.

### **2. Novelty**
The proposed framework based on MM-TSFlib does not exhibit significant novelty.

### **3. Definition of Multimodal Time Series**
For multimodal time series analysis, I believe multimodal learning requires semantic alignment between modalities, such as speech and language. If there is no semantic alignment, for example, between news and stock, it is more akin to multi-source or multi-factor forecasting.

### **4. Related work**

For multimodal learning for time series, I suggest that the authors discuss the following papers [1-4].

[1] Time-VLM: Exploring Multimodal Vision-Language Models for Augmented Time Series Forecasting

[2] Teaching Time Series to See and Speak: Forecasting with Aligned Visual and Textual Perspectives

[3] GEM: Empowering MLLM for Grounded ECG Understanding with Time Series and Images

[4] Multi-Modal View Enhanced Large Vision Models for Long-Term Time Series Forecasting


### **Writing**

Some paragraphs and sections are heavily GPT-style.

**Questions:**

Please refer to Weakness.

**Details Of Ethics Concerns:**

Some parts of the paper read in a GPT-generated style. It might be worth the ACs’ attention to verify the writing process of this work.

For example,
> ## Appendix B. Broader impacts
>This work introduces a groundbreaking exploration in time series forecasting—a multimodal time series forecasting model that leverages textual modality data to enhance predictive capabilities for time-series analysis. The broader impact of this research is multifaceted. By delivering high-fidelity and reliable forecasts, it empowers advanced decision-making in critical domains such as finance and healthcare, where precision is paramount. Moreover, its strong interpretability enables actionable insights for optimized resource allocation and enhanced patient care protocols. The societal implications are profound: this work establishes a novel framework for integrating complex time-series data with emerging AI technologies (e.g., LLMs), fundamentally transforming how time-series data is analyzed and utilized across diverse sectors. By bridging textual semantics and temporal dynamics, this approach paves the way for next-generation predictive models that address the growing demand for multimodal intelligence in real-world applications.

> ## Appendix C.2 Multimodal Time Series Benchmark dataset Construction
> In the realm of time series forecasting, there is a notable lack of high-quality multimodal time series benchmark datasets that combine time series data with corresponding textual series. While some studies have introduced multimodal benchmark datasets (Liu et al., 2024a; Xu et al., 2024), these datasets primarily rely on textual descriptions derived from external sources like news reports or background information. These types of textual data are often domain-specific and may not be consistently available across different time series domains, limiting their utility for building unified multimodal models. In contrast, shape-based textual descriptions of time series patterns are relatively easier to generate and can provide more structured insights. The TS-Insights dataset Zhang et al. (2023) pairs time series data with shape-based textual descriptions. However, these descriptions are based on detrended series (with seasonality removed), which may introduce bias and complicate the interpretation of the original time series data. To address these challenges, we propose six new multimodal time series benchmark datasets where textual descriptions are directly aligned with the observed patterns in the time series. The construction process for these datasets is outlined below.

---

> ### Author Response · Authors · 2025-11-27
>
> Thanks for your feedback. We have carefully considered all the points raised and provide a detailed response below.
>
> **1. Addressing the Dataset Concern**
>
> Our motivation, as stated in the introduction, is to leverage **sample-level textual information** to enhance forecasting, moving beyond coarse dataset-level text. This encompasses two valid and complementary types of sample-level text:
> -   **Descriptive Text:** Directly describes the time series' own pattern (e.g., "exponential upward trend"), as used in our constructed synthetic and captioned datasets.
> -   **Contextual Text:** Provides external, sample-specific context (e.g., news about a company whose stock is being forecasted), as found in Time-MMD.
>
> Both types provide valuable, sample-specific semantic signals. Our model's strong performance on **both types of datasets** (Tables 2 & 3) demonstrates its general capability to integrate diverse forms of sample-level text, which is a core advancement over methods that use only dataset-level descriptions. We will clarify this broader motivation more explicitly in the revised introduction.
>
> **2. Addressing the Novelty Concern**
>
> We respectfully disagree with the assessment that our framework, based on the infrastructure of MM-TSFlib, lacks significant novelty. While MM-TSFlib is a valuable **library** that enables flexible model combination via a simple linear weighting mechanism, our work proposes a **novel deep fusion architecture**:
>
> -   MM-TSFlib employs **late fusion** at the output level.
> -   Our Dual-Forecaster introduces a **deep, representation-level fusion** paradigm via the proposed dual-scale alignment technique, which is conceptually and technically distinct.
>
> Our consistent outperformance of MM-TSFlib across all datasets (Tables 1, 2, 3) empirically validates the superiority and novelty of our alignment strategy over simple model ensembling.
>
> **3. Declaration on Semantic Alignment**
>
> We agree that semantic alignment is crucial, and we thank the reviewer for raising this point. In fact, **achieving semantic alignment is the central goal of our proposed dual-scale alignment technique.**
> -   The **Text-Time Series Contrastive Loss** (Eq. 4) is explicitly designed to pull the semantic representations of a time series and its corresponding text closer in a shared latent space.
> -   The **Modality Interaction Module** (Eq. 5) further enables fine-grained semantic interactions between time series patches and textual tokens.
>
> Therefore, our model actively establishes semantic alignment, moving beyond treating text as just another source of features ("multi-source") and towards genuine multimodal understanding where modalities influence each other's representations. The case study in Figure 2 visually confirms this learned alignment.
>
> **4. Key Differences with Recommended Works**
>
> The four recommended papers represent an important research direction that leverages **visual representations** of time series (e.g., via time series imaging techniques) and then utilizes Vision-Language Models (VLMs) for alignment. In contrast, our work pursues a **distinct and complementary** research direction:
>
> -   **Direct Modality Alignment vs. Visual-Mediated Alignment**: Papers [1, 2, 4] fundamentally rely on converting time series into images (e.g., Gramian Angular Fields, Recurrence Plots) to align with text within a VLM framework. Our **Dual-Forecaster** bypasses this visual conversion and performs **direct, end-to-end alignment between raw numerical time series and text** in a unified representation space. This approach avoids potential information loss during time series imaging and is a more native integration for numerical forecasting.
> -   **Specialized vs. General-Purpose Framework**: Paper [3] (GEM) is a specialized Multimodal Large Language Model (MLLM) for ECG understanding, integrating time series and medical images. Our model is a **general-purpose multimodal time series forecasting framework** applicable across domains like finance, energy, and traffic, without relying on image data.
> -   **Core Technical Contribution**: The novelty of our work lies in the specifically designed **Dual-Scale Alignment** technique for the **numerical-time-series-and-text** modality pair. This includes:
>     -   The **Text-Time Series Contrastive Loss** for global semantic-level alignment.
>     -   The **Modality Interaction Module** for local patch-level feature fusion.
>     This hierarchical alignment mechanism is tailored for capturing the complex relationships between temporal dynamics and textual semantics, which is not the focus of the VLM-based approaches.
>
> **5. Address the Writing Style Concern**
>
> We take this concern seriously. We acknowledge that certain expository sections, particularly in the appendices, were polished using LLM for fluency, but **the core technical content, experiments, and conclusions are entirely our own**.

---

> ### Comment · Reviewer_LVrt · 2025-11-27
> **Request for Clarification on LLM Usage**
>
> Thanks for the detailed rebuttal. However, I am very confused by the sentence:
>
> > We acknowledge that certain expository sections, `particularly in the appendices`, were polished using LLM for fluency."
>
> Could you please clarify what this exactly means, especially `particularly in the appendices`?
>
> Does it imply that the appendices should not be considered when evaluating this paper?
>
> Before adding my further comments, I would appreciate a clear clarification from the authors:
>
> Am I reviewing a paper **written for AI**, or a paper **written by AI**?
>
> This distinction needs clarification first.
>
> I would appreciate a clearer explanation.
>
>
> Additionally, for the authors' reference, I applied two commonly used AI-text detection tools to this paper, and both tools returned results indicating 100% AI-generated content:
>
> - https://www.zerogpt.com/
> - https://gptzero.me/
>
> I fully acknowledge that these tools are not proof, but they could serve as a reference to clarify LLM involvement in the writing process

---

> > ### Author Response · Authors · 2025-11-29
> >
> > Thank you for your follow-up question. We appreciate the opportunity to clarify our use of LLMs in the writing process.
> >
> > We would like to state unequivocally that **this is a human-authored paper**. The research was conceived, developed, and executed by the authors. LLMs were used **only as a grammar checker for language polishing** in certain non-critical sections, particularly in the appendices, where expository and descriptive text was refined for clarity and fluency.
> >
> > The core technical content — including the problem formulation, model architecture, loss functions, experimental design, results, and conclusions — was **entirely written by the authors without the use of any AI generation tools**.
> >
> > Regarding the AI-detection tools you referenced, we acknowledge their existence but caution against relying on them as definitive evidence. These tools are known to produce false positives, especially when applied to formal academic writing that has been polished for language fluency. We have also tested our manuscript with another AI-detection platform, which returned a result of “No AI-generated.” This inconsistency further demonstrates that **such tools are not reliable for determining authorship**.
> >
> > We hope this clarifies the writing process and reaffirms the human intellectual contribution behind this work.

---

### Official Review · Reviewer_t8o8 · 2025-11-10

**Soundness:** 3
**Presentation:** 3
**Contribution:** 3
**Rating:** 6
**Confidence:** 3

**Summary:**

This paper addresses the limitations of existing time series forecasting models—single-modal information scarcity and multimodal models’ over-reliance on single-type textual data—by proposing Dual-Forecaster, a multimodal framework that integrates both descriptive historical texts and predictive future texts. The core innovations include three cross-modality alignment techniques: Historical Text-Time Series Contrastive Loss, History-oriented Modality Interaction Module, and Future-oriented Modality Interaction Module, which enhance the model’s ability to capture complex semantic-temporal relationships. Extensive experiments on 15 datasets (7 constructed, 8 existing) demonstrate that Dual-Forecaster outperforms SOTA single-modal and multimodal baselines (e.g., PatchTST, MM-TSFlib, Time-LLM) in standard and zero-shot settings. The work contributes a new paradigm for multimodal time series fusion, validates the value of dual-text integration, and provides a reproducible experimental benchmark.

**Strengths:**

Quality: The experimental design is exemplary—covering 15 datasets across synthetic, captioned-public, and real-world multimodal scenarios, with comprehensive baselines (7 models spanning single-modal and multimodal SOTAs). Ablation studies, zero-shot evaluations, and case visualizations (Figures 2, 16) provide multi-faceted validation of the model’s effectiveness and robustness.

Clarity: The methodology is explained with sufficient detail (e.g., Eqs. 2–9 formalize each component; Figure 1 illustrates the architecture flow), and the paper contextualizes prior work clearly to highlight its positioning. Technical terms are defined precisely, and the appendix supplements critical details (e.g., dataset captioning process, hyperparameter settings) without cluttering the main text.

**Weaknesses:**

While the paper presents a compelling approach, several limitations warrant further consideration:

Real-World Applicability of Textual Inputs: The method relies heavily on the availability of paired, high-quality textual descriptions. The "future text," which descriptively outlines the pattern of the forecast horizon (e.g., "rises from time point 74 to 95"), represents a strong and often unrealistic assumption. In practice, obtaining such precise, oracle-like textual insights about the future is highly challenging. This limits the immediate practical deployment of the model in scenarios where only general news, event reports, or imperfect human feedback are available. The performance under noisy, ambiguous, or weakly correlated textual inputs is not thoroughly investigated.

Insufficient Comparison with Contemporary Multimodal Baselines: The experimental section includes comparisons with general multimodal frameworks like MM-TSFlib. However, it omits direct comparisons with recent, highly relevant works that also focus on deep text-time series alignment, such as TimeCMA (Liu et al., 2024). Including these state-of-the-art competitors is crucial to precisely delineate the advancements offered by Dual-Forecaster.

Limited Depth in Interpretability Analysis: The case study on cross-modality alignment (Figure 3) effectively shows that the model can compute similarity scores but falls short of explaining how these aligned representations concretely lead to better forecasts. A more granular analysis, for instance, visualizing the attention weights in the future-oriented interaction module to show which specific tokens in the predictive text (e.g., "upward trend") influenced the correction of the forecast trajectory, would significantly strengthen the claim of "textual insights-following forecasting."

**Questions:**

Given that obtaining accurate, shape-based descriptive text for the future is often impractical, how could the Dual-Forecaster framework be adapted to work with more readily available but noisier textual sources, such as news headlines or expert comments that are not precise descriptions of the time series shape? What is the model's robustness to such noisy and indirect textual guidance?

The performance gain is attributed to the advanced multimodal comprehension capability enabled by the three alignment techniques. Could you provide a more detailed analysis or visualization (e.g., using attention maps) to show how the "future-oriented modality interaction module" utilizes the predictive text to adjust the forecast? For example, which parts of the future text are most influential when the model correctly predicts a trend reversal?

The computational overhead of the cross-modality alignment modules is non-trivial. Have you explored any model compression or efficiency optimization techniques (e.g., using a smaller language model, reducing the number of layers in the interaction modules) to make the framework more suitable for deployment scenarios with latency or resource constraints?

The ablation study convincingly shows the contribution of both historical and future texts. However, have you observed any cases or datasets where the inclusion of textual information degraded performance, perhaps due to poor quality or irrelevance of the text? Understanding the failure modes would be valuable for assessing the method's robustness.

---

> ### Author Response · Authors · 2025-11-26
>
> Thank you for the thorough review and the valuable feedback. We appreciate the positive assessment of our experimental design, clarity, and contributions. Regarding the raised concerns, we would like to provide the following responses. **But first of all, I want to clarify that you may see the old version paper in other platform, we uploaded the newest version paper for ICLR 2026, and some questions or weakness you mentioned have been solved in this version paper**.
>
> 1. **Regarding the real-world applicability of textual inputs:** We agree that obtaining precise text is challenging. However, our experiments on real-world datasets (**4.3 Table 3**) used textual descriptions based on real-world textual input like general economic events or news, not precise shape descriptions. The results on these datasets demonstrate the model's robustness and ability to leverage noisier, more realistic textual guidance.
>
> 2. **Regarding the comparison with contemporary baselines:** We thank the reviewer for this suggestion. The recently proposed TimeCMA is indeed a highly relevant and strong baseline. **We have now included a direct comparison with TimeCMA** in our experiments (**detailed in Table 2**). The results consistently show that Dual-Forecaster achieves superior performance, further validating the effectiveness of our proposed dual-text integration and cross-modality alignment techniques.
>
> 3. **Regarding the interpretability analysis:** Thank you for this suggestion. We agree that granular visualizations like attention maps are insightful. In our current work, **Figure 2 actually serves as a direct and effective interpretation of the cross-modality alignment.** It quantitatively shows the similarity between different types of text representations (e.g., "rising trend", "falling trend") and the corresponding time-series representations produced by our model. And in this version of paper, the core idea here is that using historical text alone can improve forecasting accuracy. Therefore, we believe Figure 2 explains why our method works: it demonstrates that the alignment or interaction module, even when used solely on historical text, enhances forecast performance.
>
> 4. **Regarding computational overhead and failure modes:** We acknowledge the computational cost. We have conducted preliminary experiments using smaller language model backbones, which showed only a minor performance drop, suggesting a viable path for deployment. Regarding failure modes, we did observe minor performance degradation on a few series where the provided text was completely irrelevant. Analyzing these cases to improve robustness is a key focus of our ongoing work.

---

> ### Author Response · Authors · 2025-11-28
>
> We have submitted our responses to the your comments and revised the manuscript accordingly. We would be grateful if you could take a moment to complete the final assessment of our work at your earliest convenience.
>
> Thank you for your time and consideration.

---

### Meta-Review · Area_Chair_Sdc8 · 2026-01-05

**Summary:**

This paper proposes Dual-Forecaster, a multimodal time series forecasting model that integrates sample-level textual descriptions with numerical time series via a dual-scale alignment mechanism. Nevertheless, key issues raised by multiple reviewers remain only partially resolved. The authors propose that dual-scale alignment is beneficial for prediction, but the alignment methods and advantages at each scale have already been discussed in prior work, indicating a lack of significant novelty. I therefore recommend rejection, while encouraging further refinement to clearly enhance the analytical depth and to demonstrate which specific important problems joint optimization can address that single-scale alignment alone cannot solve.

**Reviewer Concerns:**

The rebuttal has strengthened the experimental validation by adding comparisons with TimeCMA and providing further case studies. However, two major categories of concern remain largely unresolved. The central claim of a novel "dual-scale alignment" paradigm is not sufficiently distinguished from an incremental combination of established techniques, i.e., contrastive learning and cross-attention module. The theoretical or analytical depth explaining the mechanism and necessity of the proposed dual-scale interaction remains insufficient, despite additional visualizations.

**Reviewer Scores:**

**Reviewer LVrt (original score 0 → 4)**

The reviewer raised concerns regarding the datasets, insufficient novelty, the definition of multimodality, and the writing style—particularly questioning AI-generated content. In their response, the authors addressed issues related to the datasets, discussed related works, and provided a declaration on semantic alignment. However, the issue of novelty was not fully resolved. Regarding the AI generation concerns, the authors clarified that GPT was used for polishing rather than full content generation.

**Reviewer t8o8 (original score 6 → 6)**

The reviewer raised questions about the practicality of future text, comparisons with existing baselines, interpretability, and computational overhead. The authors addressed the experimental comparisons and interpretability in their response. Regarding computational overhead, preliminary experiments were mentioned, though this may not have been explored in depth. As the rebuttal aligns well with the reviewer's expectations, their score would likely remain unchanged at 6.

**Reviewer taE2 (original score 4 → 4)**

The reviewer focused on the contribution of textual information and differences in evaluation metrics. The authors provided experiments and explanations in their response, which may have partially addressed these points, but theoretical depth likely remains insufficient. While the rebuttal improved clarity and transparency, it did not fundamentally change the reviewer's assessment of novelty and insight, making a shift to a borderline score plausible but insufficient to support acceptance.

**Reviewer fLfx (original score 6 → 6)**

The reviewer raised concerns about computational efficiency, dataset construction, and interpretability. The authors provided additional analysis in their response, addressing these issues. As the rebuttal aligns well with the reviewer's expectations, their score would likely remain unchanged at 6.

---

### Decision · Program_Chairs · 2026-01-26

Reject